# Applications of a disintegration transformation

**Praveen Narayanan**
Indiana University

**Chung-chieh Shan**
Indiana University

## Abstract

We describe examples of applying a disintegration transformation on probabilistic programs to obtain posterior distributions, calculate symbolic representations of densities, and generate Markov Chain Monte Carlo (MCMC) samplers.

Conditioning and density are critical components of probabilistic inference. The measure-theoretic notion of disintegration generalizes both conditioning and density (Chang and Pollard, 1997), and it can be implemented as a source-to-source transformation on probabilistic programs (Shan and Ramsey, 2017). In this work we describe example applications of such a disintegration transformation.

Disintegration may be implemented to condition first-order probabilistic programs that apply piecewise-invertible mathematical operations to observable expressions. Such a transformation can be formally verified given a semantics of $s$-finite kernels (Staton, 2017) and a calculus of distributions based on this semantics. Given a language with primitives for manipulating stochastic arrays, a disintegration technique can be improved to scale efficiently over arbitrary numbers of observations (Narayanan and Shan, 2017). Furthermore, the transformation can be implemented to correctly address programs over sophisticated measure spaces, including discrete-continuous mixtures, disjoint sums, and dependent products (Narayanan and Shan, 2019).

Our first example is that of using disintegration to condition a Gaussian mixture model. Here by conditioning we mean producing a program representation of a posterior distribution given a program representation of the model and a term (expression) representing our observation(s). We illustrate the output of our conditioning tool on a mixture model that observes an array of labeled real numbers arising from a mixture of an arbitrary number of Gaussian distributions.

Next we use our disintegration transformation to implement a generalized density tool more powerful (in terms of distribution types) than existing density calculation techniques. We illustrate this tool on an application of mutual information estimation (Gao et al., 2017), showing that it can produce exact representations of the density between discrete-continuous mixture distributions.

Finally, we describe how to automatically generate a single-site Metropolis-Hastings MCMC sampler (as a probabilistic program), an application that calls for using our disintegration transformation both as a conditioning tool as well as a density calculator. First, we use the conditioning tool to obtain the target distribution for MCMC sampling. Second, we use the density tool to calculate the acceptance ratio for a measure-theoretic Metropolis-Hastings-Green transformation (Green, 1995). This is a transformation in that it takes as input and produces as output probabilistic programs.

## 1 Conditioning a Gaussian mixture model

We start by using a disintegration transformation to perform conditioning, i.e., to go from a Bayesian model to a posterior distribution, both represented as probabilistic programs in the same language. The language we use in this work is Hakaru, which is strongly-typed and first-order, and in which programs are written in a monadic style and denote $s$-finite kernels.

In the center of Figure 1 we show a program written in Hakaru (Narayanan et al., 2016). Named $gmm$, it represents a Gaussian mixture model over $N$ real numbers drawn from $K$ Gaussian clusters.

Preprint. Under review.

$$gmm : \quad \mathbb{N} \to \mathbb{N} \to \mathbb{M}\left((\mathbb{A}\,\mathbb{R} \times \mathbb{A}\,\mathbb{N}) \times \mathbb{A}\,\mathbb{R}\right)$$

$gmm = \lambda K.\,\lambda N.$
    **do** $\{\vec{\theta} \leftsquigarrow$ **dirichlet**(**array** $K$ _. 1);
        $\vec{\mu} \leftsquigarrow$ **plate** $K$ _.
            **normal** $0\ 0.01;$
        $\vec{\sigma} \leftsquigarrow$ **plate** $K$ _.
            **gamma** $2\ 0.05;$
        $\vec{z} \leftsquigarrow$ **plate** $N$ _.
            **categorical** $\vec{\theta};$
        $\vec{x} \leftsquigarrow$ **plate** $N\,i.$
            **normal** $\vec{\mu}[\vec{z}[i]]\ \vec{\sigma}[\vec{z}[i]]$
    **return** $((\vec{x}, \vec{z}), \vec{\mu})\}$

$disintegrate\ (gmm\ K\ N)\ (t_x, t_z)$
$\Longrightarrow$ **do** $\{\vec{\theta} \leftsquigarrow$ **dirichlet**(**array** $K$ _. 1);
    $\vec{\mu} \leftsquigarrow$ **plate** $K$ _. **normal** $0\ 0.01;$
    $\vec{\sigma} \leftsquigarrow$ **plate** $K$ _. **gamma** $2\ 0.05;$
    _ $\leftsquigarrow$ **plate** $N\,i.$
        $(\vec{\theta}[t_z[i]] \div sum\ \vec{\theta})$
        $\odot$ **return** $();$
    _ $\leftsquigarrow$ **plate** $N\,i.$
        **dnorm** $\vec{\mu}[t_z[i]]\ \vec{\sigma}[t_z[i]]\ t_x[i]$
        $\odot$ **return** $();$
    **return** $\vec{\mu}\}$
  : $\mathbb{M}\,(\mathbb{A}\,\mathbb{R})$

Figure 1: A Bayes net for a Gaussian mixture model (bottom-left), the Hakaru program $gmm$ (center), and the disintegration of $gmm\ K\ N$ at the observed point $(t_x, t_z)$ (right)

The first line of this program is its type, a function from two natural numbers ($K$ and $N$) to a *measure* ($\mathbb{M}$) over pairs containing arrays ($\mathbb{A}$). The program $gmm$ is written in a monadic sequence of statements that *bind* random variables to distributions. We distinguish the notation of random variables that represent arrays, such as $\vec{\theta}$ bound to **dirichlet**, from that of variables representing scalar values, such as $x$ bound to **normal**. The **plate** construct expresses repeated independent sampling and returns an array. This program overall represents a joint distribution over two dimensions. In one dimension, each element pairs an array of reals ($\vec{x}$, points drawn from the mixture) with an array of natural numbers ($\vec{z}$, the cluster labels for those points). In the other dimension, each element is an array of reals ($\vec{\mu}$, the cluster centers).

To condition this model we use the *disintegrate* transformation of type

$$disintegrate : \mathbb{M}\,(\alpha \times \beta) \to \alpha \to \{\mathbb{M}\,\beta\}. \tag{1}$$

This type informs us that *disintegrate* transforms a joint distribution $\mathbb{M}\,(\alpha \times \beta)$ into a set of conditional distributions $\{\mathbb{M}\,\beta\}$ given the observed expression $\alpha$. This set may be empty to account for failure. In the case of $gmm$, $\alpha = \mathbb{A}\,\mathbb{R} \times \mathbb{A}\,\mathbb{N}$ and $\beta = \mathbb{A}\,\mathbb{R}$.

On the right-hand-side of Figure 1 we show the result of using *disintegrate* to condition $gmm$ with a symbolic observation term $(t_x, t_z)$. Each unobserved term from the input model remains untouched in the output program. For the observed terms, here $\vec{x}$ and $\vec{z}$, the binding distributions get converted into densities. Here we use the $\odot$ operator (found in other languages as *factor* or *score*) to re-weight or scale a distribution by a non-negative number (of type $\mathbb{R}_+$). The **dnorm** construct is syntactic sugar for the density of **normal** with respect to Lebesgue. This output program of type $\mathbb{M}\,(\mathbb{A}\,\mathbb{R})$ represents the unnormalized posterior distribution, which *disintegrate* can produce without unrolling the arrays in the input. The process takes time linear in the number of indices needed to select any array element (which in this case is 1).

Disintegration can be used in this manner to transform joint measures into unnormalized posterior measures for a large class of probabilistic programs. It effectively handles programs whose observed (or conditioned) sub-parts arise from distributions over arrays, discrete-continuous mixture spaces, disjoint sums, and dependent products.

## 2 Calculating density to estimate mutual information

A disintegration transformation can also be used to build density calculation tools. In this section we define two density calculators, classify them as *restricted* and *unrestricted* in terms of the *base measures* with respect to which they operate, and apply the latter tool to estimate mutual information between pairs of random variables.

To build the *restricted* density calculator, we use the typical understanding of the density of distribution $\mathbb{M}\,\alpha$ as a $\alpha \to \mathbb{R}_+$ function. Here our density tool becomes a special case of *disintegrate*:

$$density : \mathbb{M}\,\alpha \to \alpha \to \{\mathbb{M}\,\mathbb{1}\} \tag{2}$$

$$density\ m = disintegrate\ (m \otimes \textbf{return}\ ()). \tag{3}$$

$$density : \mathbb{M}\,\alpha \to \mathbb{M}\,\alpha \to \alpha \to \{\mathbb{R}_+\}$$
$$density\;\mu\;\nu\;t = \text{let } \mu' = \mu \otimes \mathbf{return}\;()$$
$$\nu' = \nu \otimes \mathbf{return}\;()$$
$$\lambda\; = \mathit{infer}\;\mu'\;t + \mathit{infer}\;\nu'\;t$$
$$\text{in } |\mathit{check}\;\mu'\;\lambda\;t| \div |\mathit{check}\;\nu'\;\lambda\;t|$$

Figure 2: Unrestricted density calculation using ratios of Radon-Nikodym derivatives

Informally, densities are disintegrations obtained by observing all dimensions of the joint measure. In order to do this correctly using $disintegrate$, we use $\otimes$ to convert a measure $m : \mathbb{M}\,\alpha$ into an equivalent measure of type $\mathbb{M}\,(\alpha \times \mathbb{1})$:

$$(\otimes) : \mathbb{M}\,\alpha \to \mathbb{M}\,\beta \to \mathbb{M}\,(\alpha \times \beta) \tag{4}$$

$$m \otimes n = \mathbf{do}\;\{x \leftarrow m;\; y \leftarrow n;\; \mathbf{return}\;(x, y)\}. \tag{5}$$

The type $\mathbb{1}$ has just one element (). The result of (3) is an $\alpha \to \{\mathbb{M}\,\mathbb{1}\}$ function since densities (just like disintegrations) can fail to exist. The $\mathbb{M}\,\mathbb{1}$ produced by disintegration is isomorphic to $\mathbb{R}_+$.

The density tool of Equation (3), however, is insufficient for applications such as *mutual information estimation* (Gao et al., 2017), or MCMC sampling (considered in the next section). Given a probability measure $\mu$ (on a space $\alpha \times \beta$) and its marginals $\mu_a$ and $\mu_b$, mutual information is defined in terms of a *Radon-Nikodym derivative* $d\mu/d\nu$ with respect to a measure $\nu = \mu_a \times \mu_b$. Gao et al. show that this derivative exists ($\mu \ll \nu$) and describe a novel algorithm to *estimate* this derivative for discrete-continuous mixture distributions. We want to calculate *an exact representation* of the derivative $d\mu/d\nu$, but $density$ from Equation (3) only takes $\mu$ as argument, remaining implicit about the *base measure* $\nu$. To calculate this derivative we need a tool that allows arbitrary Hakaru measures as base measures, i.e., a tool of type $\mathbb{M}\,\alpha \to \mathbb{M}\,\alpha \to \alpha \to \{\mathbb{R}_+\}$.

To build this *unrestricted* density calculator, we choose a measure—such as $\lambda = \mu + \nu$—with respect to which both $\mu$ and $\nu$ have Radon-Nikodym derivatives, then calculate the derivative $d\mu/d\nu$ as a ratio of two derivatives, i.e., by dividing two real numbers (Geyer, 2011, p. 40, eq. 1.26):

$$\frac{d\mu}{d\nu} = \frac{d\mu/d\lambda}{d\nu/d\lambda}. \tag{6}$$

To obtain *common* base measures such as $\lambda$, we can use an $infer$ routine found within $disintegrate$:

$$disintegrate : \mathbb{M}\,(\alpha \times \beta) \to \alpha \to \{\mathbb{M}\,\beta\} \tag{7}$$

$$disintegrate\;m\;t = \text{let } b = \mathit{infer}\;m\;t \text{ in } \mathit{check}\;m\;b\;t \tag{8}$$

We see that disintegration is composed of two tools, one that *infers* a base measure for an input joint measure, and another that *checks* the disintegration of an input joint measure with respect to a given base measure.

$$\mathit{infer} : \mathbb{M}\,(\alpha \times \beta) \to \alpha \to \mathbb{B}\,\alpha \tag{9}$$

$$\mathit{check} : \mathbb{M}\,(\alpha \times \beta) \to \mathbb{B}\,\alpha \to \alpha \to \mathbb{M}\,\beta \tag{10}$$

The type $\mathbb{B}$ represents base measures, and it is a subset of the type $\mathbb{M}$ of measures expressible in full Hakaru. The base measures are implicit in the types of $density$ and $disintegrate$. Their restricted language makes it possible to define tools such as $infer$, and it is by extending this language that we are able to handle the large class of probabilistic programs mentioned in the previous section.

In Figure 2 we show how $infer$ can be used to obtain the common base measure $\lambda$ and build the unrestricted density tool. We sum two base measures using a $+$ operator that acts as a join on the density preorder. Since $\lambda$ remains in the restricted language $\mathbb{B}\,\alpha$, we may use $check$ to calculate two separate Radon-Nikodym derivatives and output their ratio. Note that we also use a *total* map $\lambda\nu.\,|\nu|$ that is easy to implement as a program transformation, though it can produce integrals and sums that witness the fundamental intractability of probabilistic inference.

With the unrestricted $density$ of Figure 2 we can now obtain an exact representation of $d\mu/d\nu$ needed for mutual information estimation:

$$d_{MI} : \mathbb{M}\,(\alpha \times \beta) \to (\alpha \times \beta) \to \{\mathbb{R}_+\} \tag{11}$$

$$d_{MI}\;\mu = density\;\mu\;(\mathit{fmap}\;\mathbf{fst}\;\mu \otimes \mathit{fmap}\;\mathbf{snd}\;\mu). \tag{12}$$

Here $\nu$ is represented as the product of marginals obtained using $fmap : (\alpha \to \beta) \to \mathbb{M}\,\alpha \to \mathbb{M}\,\beta$.

For example, here is the model defined by by Gao et al. (2017) as *Experiment I*, modified with a simpler covariance matrix for the two-dimensional normal distribution.

$$\xi \;: \mathbb{M}\,(\mathbb{R} \times \mathbb{R}) \tag{13}$$

$$\xi = (\mathbf{normal}\,0\,1 \otimes \mathbf{normal}\,0\,1) \tag{14}$$
$$\quad \oplus \mathbf{atoms}\,[(0.45,(1,1)),(0.45,(-1,-1)),(0.05,(1,-1)),(0.05,(-1,1))]$$

This is a discrete-continuous mixture of a two-dimensional normal distribution and weighted point-mass distributions at atoms such as $(1,1)$ and $(-1,1)$. The mutual information density transformation of Equation (12) produces exact values for this mixture; for example, the unnormalized density of $\xi$ at the point $(1,1)$ is $9/5$.

## 3   Generating a Metropolis-Hastings-Green MCMC sampler

As our final application, we use $density$ from Figure 2 to build an MCMC transformation. This tool takes two probabilistic programs as input—a *target* distribution $\eta$ and a *proposal kernel* $Q$—and produces an output program that when run can produce a sample for the Markov chain.

$$mhg : \mathbb{M}\,\alpha \to (\alpha \to \mathbb{M}\,\alpha) \to \alpha \to \mathbb{M}\,\alpha$$
$$mhg\,\eta\,Q\,x = \mathbf{do}\;\{y \leftsquigarrow Q\,x;$$
$$\qquad\qquad\qquad r_{xy} \leftsquigarrow \mathbf{return}\,(greenRatio\,\eta\,Q\,(x,y));$$
$$\qquad\qquad\qquad a_{xy} \leftsquigarrow \mathbf{return}\,\min(1, r_{xy});$$
$$\qquad\qquad\qquad b \leftsquigarrow \mathbf{bern}\,a_{xy};$$
$$\qquad\qquad\qquad \mathbf{return}\,(\mathbf{if}\;b\;\mathbf{then}\;y\;\mathbf{else}\;x)\}$$

Figure 3: A program transformation implementing the Metropolis-Hastings-Green update

We show this transformation in Figure 3. It implements the Metropolis-Hastings-Green (MHG) update (Geyer, 2011; Green, 1995; Tierney, 1998), which is more general than the similarly structured Metropolis-Hastings (MH) update (Hastings, 1970; Metropolis et al., 1953) due to a Radon-Nikodym derivative between two joint measures. We use $density$ to calculate this derivative, in a tool which we follow Geyer in naming $greenRatio$.

$$greenRatio : \mathbb{M}\,\alpha \to (\alpha \to \mathbb{M}\,\alpha) \to (\alpha \times \alpha) \to \{\mathbb{R}_+\}$$
$$greenRatio\,\eta\,Q = density\,(fmap\,switch\,(\eta \otimes\!\!= Q))\,(\eta \otimes\!\!= Q)$$

$$switch : (\alpha \times \beta) \to (\beta \times \alpha) \qquad\qquad (\otimes\!\!=) : \mathbb{M}\,\alpha \to (\alpha \to \mathbb{M}\,\beta) \to \mathbb{M}\,(\alpha \times \beta)$$
$$switch\,p = (\mathrm{snd}\,p, \mathrm{fst}\,p) \qquad\qquad m \otimes\!\!= k = \mathbf{do}\;\{x \leftsquigarrow m;\,y \leftsquigarrow k\,x;\,\mathbf{return}\,(x,y)\}$$

Figure 4: The Green ratio used to calculate the acceptance probability of an MHG update

Figure 4 illustrates $greenRatio$. The first two arguments to $greenRatio$ are the target distribution and proposal kernel for MCMC sampling. The target is typically a posterior distribution, such as the one on the right-hand-side of Figure 1. The proposal kernel can be any Hakaru program describing an MCMC search strategy; for example, here is a *single-site* proposal scheme (Geyer, 2011; Wingate, Stuhlmüller, and Goodman, 2011) for when $K = 2$:

$$Q_{ss} = \lambda(x_1, x_2).\quad \mathbf{do}\;\{x_1' \leftsquigarrow \mathbf{normal}\,x_1\,0.1;\,\mathbf{return}(x_1', x_2)\} \tag{15}$$
$$\qquad\quad \oplus \mathbf{do}\;\{x_2' \leftsquigarrow \mathbf{normal}\,x_2\,0.1;\,\mathbf{return}(x_1, x_2')\} : \mathbb{R}^2 \to \mathbb{M}\,\mathbb{R}^2$$

We emphasize that $greenRatio$ produces syntax, i.e., an expression (of type $\mathbb{R}_+$) for the Radon-Nikodym derivative at a point $(x,y)$. For example, when $(x,y) = ((0.37, 0.42), (0.37, 0.19))$, $greenRatio$ evaluates to the expression $\exp(-23/240)$ given $Q_{ss}$ and $gmm$ conditioned on $(t_x, t_z) = ((0.1, 0.2), (0, 1))$. Similarly, $mhg$ produces code that may be used, for instance, to chain samples together or to compose *mixtures* and *cycles* of MCMC kernels (Andrieu et al., 2003).

The generality of disintegration facilitates reusable program transformations. One kind of reusability is within a single inference pipeline: we showed how $disintegrate$ can be used for producing the target (via conditioning) as well as the acceptance ratio (via density calculation). Another kind is reusability across general state spaces. For instance, $mhg$ may easily be used with *reversible-jump* proposal kernels (Richardson and Green, 1997) when $\alpha$ is a disjoint sum space.

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
