# OpenReview forum: "Applications of a disintegration transformation"
_NeurIPS.cc/2019/Workshop/Program_Transformations — Program Transformations @NeurIPS2019 Oral_

### Official Review · AnonReviewer1 · 2019-09-26
**Interesting and relevant work on a program transformation**

**Confidence:** 4
**Rating:** 8

**Review:**

This work describes the use of the 'disintegration transform' for the transformation and manipulation of probabilistic programs. I found the work dense, but interesting to read. Overall, I think this seems like a very interesting and relevant work for our workshop. I would make a few suggestions for the authors:
- Define 'disintegration transform' more clearly and early in the paper. I saw lots of cool things you could do with this transform, but I still don't really know how it works. If it's a fundamental program transformation akin to autodiff, shout that from the roof tops.
- Highlight the language you've implemented this in, Hakaru, more prominently. Having a real system people can use to try out your ideas is very powerful. I would have liked to see a small example program on the first page.

Overall, the paper itself could use improvements in readability and exposition, but we should accept it and hear more about the work from the authors.

---

### Official Review · AnonReviewer2 · 2019-09-30
**Interesting representation and transformations of probabilistic programs**

**Confidence:** 3
**Rating:** 8

**Review:**

The submission presents a few examples, particularly relevant for machine learning and statistics, of what can be achieved using a disintegration transformation, and related constructs (infer, check).
Notable points are:
- The ability to represent and transform operations on array (rather than only scalars) in parallel, without unrolling or adding dependencies between elements.
- Support for discrete-continuous mixtures
- The versatility of the transformations

The first point is especially relevant in the era of parallel hardware and large amount of data (real or simulated).

If selected as an oral, I would suggest spending a bit more time on the "usual" constructs and syntax of PPLs, which may not be known by a machine learning audience, to emphasize what is new in this line of work. An intro on disintegration may also be useful if feasible.

---

### Decision · Program_Chairs · 2019-10-01

**Decision:**

Accept (Oral)

**Comment:**

The reviewers argued this is good work which is squarely in the scope of the workshop, warranting an oral presentation. However, both reviewers noted problems with the presentation (readability issues, densely written, hard to follow for an outside audience). Considering the interdisciplinary nature of this workshop, we suggest the authors focus on making their work accessible to a broad audience when preparing their oral presentation.